# "The child of your fellow is your child": Building on existing protective norms to engage men as caregivers; qualitative findings from an exploratory evaluation of an edutainment intervention to prevent age-disparate transactional sex

Alicia Sharif[1]◎, Marjorie Pichon[1]◎*, Veronicah Gimunta[2], Oscar Rutenge[3], Revocatus Sono[2], Ana Maria Buller[1]‡, Lottie Howard-Merrill[1]¤‡

1 Gender Violence & Health Centre, Department of Global Health and Development, London School of Hygiene & Tropical Medicine, London, United Kingdom, 2 Amani Girls Organization, Mwanza, United Republic of Tanzania, 3 Tanganyika Christian Refugee Service, Dar es Salaam, United Republic of Tanzania

◎ These authors contributed equally to this work.
‡ AMB and LH-M also contributed equally to this work.
¤ Current address: Department of Education, Practice and Society, Institute of Education, University College London, London, United Kingdom
* Marjorie.Pichon@lshtm.ac.uk

## Abstract

Evidence on the importance of engaging men in preventing violence against women and girls has increased over the past decade, yet efforts often focus on men as partners rather than caregivers. This paper examines qualitative data from an evaluation of the Learning Initiative on Norms, Exploitation and Abuse radio drama intervention aimed at preventing age-disparate transactional sex in Tanzania's Shinyanga region. We delivered the drama to households on flash drives and led structured household discussion sessions. We conducted in-depth interviews with 18 adult men caregivers before (September 2021) and after the intervention (December 2021) and used thematic analysis and the framework method to examine indications of change in attitudes, beliefs, norms and behaviours. Findings provided practical lessons for future interventions aiming to engage men, specifically as caregivers, and demonstrated promising indications of change. We found that the home-based delivery of the intervention coupled with household discussions led to high engagement from men. This engagement fostered expansion of participant's conceptualisation of fatherhood to include discussing (age-disparate transactional) sex with adolescent girls, as well as with other men in the community, with the goal of protecting girls. We also found that the drama led to an expansion of the existing norm that it is everyone's responsibility within a community to protect girls, to include protecting girls from age-disparate transactional sex; highlighting the success of norms programming that reinforces

**Data availability statement:** The research is based upon an analysis of in-depth interviews conducted with a small sample of adolescent girls and caregivers. These transcripts cover topics that are considered sensitive by participants and contain context-specific information that would enable them to be identified if transcripts were made available in their entirety. To safeguard the confidentiality and welfare of the individuals interviewed, we are therefore only able to share excerpts of anonymised transcripts that underpin the conclusions drawn in our manuscript. These excerpts may be requested for use in ethically approved research via the LSHTM Data Compass repository at https://doi.org/10.17037/DATA.00003750 or by emailing researchdatamanagement@lshtm.ac.uk.

**Funding:** This research was funded by the OAK Foundation: https://oakfnd.org/ (Grant No. OFIL-20-236, recipient AMB), Wellspring Foundation: https://thewellspringfoundation.org/ (Grant No. 13343, recipient AMB) and FELM (Finnish Evangelical Lutheran Mission): https://felm.org/ (Grant No. TZ710, recipient AMB). The funders had no role in study design, data collection and analysis, decision to publish, or preparation of the manuscript.

**Competing interests:** The authors declare that no competing interests exist.

existing positive norms rather than introducing new norms. This study highlights the potential for edutainment using a variety of acceptable and relatable characters and storylines to model positive behaviours performed by men, and home-based, discussion-oriented approaches to foster meaningful change. Finally, this study's findings offer valuable insights for developing effective strategies to engage men in violence prevention, while ensuring they remain accountable to the needs and priorities of women and girls.

## Introduction

### Engaging men to prevent violence against women and girls

Over the past decade the importance of engaging men and boys in efforts to prevent and reduce violence against women and girls (VAWG) has been well-established, with 'engaging men' increasingly included as a component in violence prevention programmes and campaigns around the world [1–3]. These efforts range from delivering interventions exclusively to men, to those involving both men and women, as well as implementing programming at the community in addition to at the individual level [4].

Despite the increasing focus of involving men and boys in VAWG prevention efforts, concerns have been raised, particularly within activist circles, that such efforts may inadvertently reinforce men's hegemony over women by directing resources and attention away from women-centred programmes [1,5]. Another concern is that the content of these programmes may further encourage men to view women primarily in relational roles – such as wives, mothers and daughters – rather than as individuals in their own right [5]. Others have also questioned the ability of programmes to actually change men's behaviours.

Evaluations of interventions aimed at engaging men to prevent VAWG have shown that whilst men's attitudes may change, structural barriers often prevent these changes from translating into behavioural change [1,3,4]. Factors such as poverty, racism and migration status can constrain men's sustained engagement in violence prevention, especially when programmes require disadvantaged men to critically evaluate their own power and privilege over others [1]. Similarly, whilst men may support the message of gender equality-based work, their willingness to undermine patriarchy by mobilising other men and advocating for wider change is often limited [6,7].

These concerns are valid, and it is important to recognise the potential for well-intentioned programmes to cause harm. This does not mean, however, that efforts to engage men and boys should be abandoned. For instance, a 2021 brief from the Sexual Violence Research Initiative, one of the leading dissemination networks in the field, emphasizes that "while women must remain at the centre of and lead efforts to prevent the violence against them, men must join the movement as allies", through accountable, sustainable and effective interventions [8]. Indeed, evidence shows that gender-transformative programmes– those seeking to transform

gender roles rather than ignore or accommodate them – are effective in improving health outcomes, including the prevention of VAWG [4,9].

Most research on engaging men in VAWG efforts has focused on men as sexual partners, such as engaging men in couples in intimate partner violence (IPV) prevention programmes (e.g. [10]). However, a growing body of research suggests encouraging men to embrace their role as caregivers can significantly improve outcomes for their children, their partners and themselves. A recent review of father-inclusive interventions on parental, couple and early childhood outcomes saw both an improvement in child outcomes, such as nutrition and language development, and improvements in parental and wider family relationship dynamics [11]. Evidence also suggests that this goes beyond early childhood outcomes – the more quality time fathers spend with their adolescent children, the lower their chances of engaging in unprotected sex, and the lower their number of sexual partners [12]. Furthermore, connectedness between fathers and their adolescent daughters can decrease risky behaviour, including criminal behaviour and using alcohol and/or drugs [13–16].

Additionally, research has found that engaging men as fathers can reduce physical and sexual violence against their partners, and physical punishment of children [17], including in Tanzania [18]. Parenting programmes have also been linked to improved parental mental health, which in turn contributes to reductions in violence and maltreatment of children and strengthened parent-child relationships [19]. Thus, fatherhood represents a crucial opportunity for intervention. Moreover, a recent meta-analysis of 27 randomised controlled trials found that interventions were equally effective at reducing IPV when targeting communities, women only, men only or couples [20], suggesting the need for programming targeting each group.

Engaging fathers in parenting programmes, however, is challenging, as highlighted by low participation rates worldwide [21,22]. Historically, this was thought to be due to men's lack of interest in caregiving [23], but evidence now points to structural, financial and norm-based barriers limiting their engagement [24]. For example, workplaces around the world offer men less flexible parental leave than women, making it harder for men to take time off for caregiving [25]. Where parental leave is available for men, it is often unpaid or paid at a lower rate than their usual work, which can make it financially unfeasible [25]. It can also be difficult for men who work long hours to travel long distances for programming [26]. This is compounded by societal norms that place caregiving responsibilities on women and financial provision on men – although there is evidence that these norms are gradually shifting [24]. Thus, programmes must be carefully designed with these challenges in mind to engage men effectively.

In Eastern Africa there has been some limited success in engaging men in parenting programmes, which provide valuable lessons. The 'Parenting for Respectability' programme in Uganda, for instance, actively included fathers by aligning its aims with their understandings of 'good' parenting – that is, raising well-behaved and respectful children [23]. While the programme achieved good engagement from fathers, social pressures to conform to conventional masculinities persisted after the intervention, highlighting the need for community-level programming as well. Moreover, there was evidence that fathers began to 'police' mothers and their behaviours towards their children, rather than working collaboratively and in a gender-equitable way, suggesting that joint, or family-wide interventions might be more effective than engaging fathers alone, and could promote men's accountability towards women and girls [27].

The 'REAL Fathers Initiative' in northern Uganda is another example which aimed to reduce children's exposure to violence, including witnessing IPV and experiencing maltreatment, through a 12-week mentoring programme and a community poster campaign [28]. Mentors modelled non-violent responses to children's behaviours to improve fathers' parenting skills. After engagement with the programme participants reported increased confidence using non-violent discipline methods and engaging in positive parenting behaviours such as reading with their child. The evaluation also found decreased levels of physical child punishment and both verbal and physical IPV at long-term follow-up. However, there was no significant effect on the gender norms underpinning IPV, likely due to the short intervention period, indicating the need for longer-term programming [28].

                                    

There have also been examples of successful parenting interventions in Tanzania. Recently, the 'Parenting for Lifelong Health' programme, which was adapted from the South African context, was found to be effective in reducing child mal-treatment, IPV, and parent and child depression, as well as improving sexual health communication between parents and daughters, among other outcomes [23]. The programme was developed to address the underlying drivers of HIV among adolescent girls through a 14-session curricula which was delivered to over 75,000 parents and adolescent girls [23]. Over one-third of the parents included were men, suggesting there is a high demand for parenting programmes from male care-givers in Tanzania. To further scale-up this programme while reducing costs, the researchers developed a phone application to deliver the material to parents on mobile phones, which were distributed throughout communities in the Mwanza region [29]. This 'ParentApp' programme is currently being evaluated through a cluster-randomised controlled trial, which is expected to come to completion in 2025 [29]. Whilst these examples offer important insights into how to better engage fathers in parenting programming, most programmes have focused on early childhood outcomes. In particular, there has been a strong focus in programming on preventing violence against children as a primary outcome. While these pro-grammes tend to utilise gender-transformative approaches, which also have the potential to impact many other outcomes, including age-disparate transactional sex, few interventions specifically target men as fathers of adolescent girls, or target male caregivers with the primary goal of improving their daughters' sexual and reproductive health outcomes.

## Age-disparate transactional sex

There is a disproportionately high incidence of HIV among adolescent girls and young women in sub-Saharan Africa, where according to a 2023 UNAIDS report, they account for 61% of those living with the virus [30]. This imbalance is partly attributed to age-disparate transactional sex, which heightens their risk of engaging in unprotected sex and experi-encing sexual violence, due to the inherent power imbalances that underpin these relationships [31]. Adolescent girls and young women involved in such relationships are not only younger than their partners but also lack access to resources [32], limiting their ability to decline sex or negotiate condom use [33].

Transactional sex is defined as "non-commercial, non-marital sexual relationships motivated by the implicit assump-tion that sex will be exchanged for material support or other benefits" [34]. Unlike commercial sex work, transactional sex involves relationships where individuals often identify as boyfriends and girlfriends without explicit negotiations of 'price' or 'services' [35]. In the global literature, *age-disparate* transactional sex refers to transactional sex relationships involving girls under 18 years with partners at least a five-years older [36], though the age gap is typically larger [37].

Research on the role of men in age-disparate transactional sex and transactional sex more broadly is growing. This literature has primarily focused on the associated risks, such as HIV and other sexually transmitted infections, as well as men's motivations for engaging in (age-disparate) transactional sex relationships [38–40], such as to fulfil the gender norms of male provision and heightened sexuality [34,38,41,42]. However, to the authors' knowledge, no studies have examined the role of men caregivers in preventing their daughters from engaging in age-disparate transactional sex. This study begins to fill this gap.

## The current study

This paper examines indications of change using qualitative data collected during a small-scale pilot evaluation of the Learning Initiative on Norms, Exploitation and Abuse (LINEA) radio drama intervention. The LINEA radio drama, co-created by the London School of Hygiene & Tropical Medicine (LSHTM), Amani Girls Organization (AGO) and Media for Development International Tanzania, was designed specifically for the Tanzanian context. It aims to shift the social norms that drive age-disparate transactional sex using an education-entertainment or 'edutainment' approach [43]. Explor-atory evaluations were carried out in the Shinyanga and Kigoma regions of Tanzania in 2021 to assess indications of change of the radio drama (for further detail see [43–45]).

We selected Kishapu district in the Shinyanga region as the study site for several reasons, including ease of imple-mentation and the prevalence of issues associated with age-disparate transactional sex. The choice was also influenced by a pre-existing relationship with the implementing partner organisation, the Tanganyika Christian Refugee Service (TCRS), which is based in the area. TCRS programming focuses primarily on building community resilience for climate change adaptation and mitigation and preventing gender-based violence. Shinyanga is a predominantly rural region where agriculture (crop cultivation and livestock rearing) is the main occupation [46]. It has higher than average rates of HIV [47] compared to the national average (4.4%) [48], and among the highest prevalence of IPV in the country [49], with estimates as high as 78% of ever-married women reporting experiencing violence in their lifetime [50]. The region also has the highest prevalence of child marriage in Tanzania, with estimates between 59–64% [50,51], and a recent study highlighting a culture of tolerance towards sexual violence among married adolescent girls [52]. However, research with caregivers and daughters has also shown strong support for parent-child connectedness as a strategy to prevent early sexual debut among adolescent girls [53].

This paper is part of a set of three focusing on these exploratory evaluations in the Shinyanga region with adolescent girls and men and women caregivers, ahead of the trial evaluating LINEA taking place in 2023–2025 in Mwanza region. The initial papers by Pichon et al. [44] and Pichon & Buller et al. [45] present findings on the impact of the LINEA drama on participants. The first focuses on changes in participants' knowledge, attitudes, beliefs and social norms linked to age-disparate transactional sex [44]; While the second paper examines changes in participants' educational aspirations and gender equitable attitudes towards work [45]. Findings from the study conducted in Kigoma region, where the radio drama was broadcast on a local radio station rather than given to participants on a flash drive, are presented in another paper by Howard-Merrill & Pichon et al. [54]. The current paper contributes to this body of research by further examin-ing qualitative data from Kishapu district, Shinyanga region. In this paper we specifically explore men's perspectives in more depth to better understand the mechanisms of their engagement with the radio drama, providing practical lessons for future programming targeting fathers and men caregivers more generally. Furthermore, by investigating indications of change in men caregivers attributed to the LINEA drama, this paper provides nuanced insights into men's roles in age-disparate transactional sex, going beyond seeing men solely as perpetrators, and exploring their potential to support girls and engage the wider community in age-disparate transactional sex prevention efforts.

## Methods

This paper uses qualitative data from baseline (September 2021) and endline (December 2021) in-depth interviews (IDIs) from a mixed-methods, exploratory evaluation of the LINEA radio drama carried out in the Kishapu district of the Shin-yanga region, in northern Tanzania [44]. We conducted interviews before and after the intervention with 18 men caregivers (total number of IDIs = 36) to enable an analysis of any self-reported and observed changes in attitudes, beliefs, norms and behaviours. While midline interviews were conducted in the original study, they were excluded from the current analy-sis because they occurred only three and a half weeks before endline interviews, and were therefore deemed to be more suitable to inform the process evaluation of the intervention rather than to assess indications of change [45].

### The LINEA radio drama and household discussion sessions

The radio drama is titled *"Msichana Wa Kati"* (*"The Girl in the Middle"*) in Kiswahili. It consists of 39 episodes between 15 and 20 minutes long and was designed to be played once per week over a period of nine months on the radio [44]. The drama encourages listeners to critically reflect on the drivers of age-disparate transactional sex, and was specifically designed to promote community-level change by including several positive role models – both male and female – who demonstrate how members of the community can help to change harmful behaviours related to age-disparate transac-tional sex.

Among other male role models within the story (described in Table 1), the character that speaks directly to male care-givers (and therefore is relevant to this paper) is Tizo, the father of Amali, who is the central adolescent girl character. At the beginning of the story, he abides by traditional, patriarchal norms; often travelling for work and barely communicating with his daughters. After an injury that leaves him unemployed and at home, he becomes more involved in his daughters' lives, going to speak to their school headmistress (Fig 1), encouraging their careers, and ultimately helping them avoid age-disparate transactional sex. He also ensures that the teacher who engages in age-disparate transactional sex with his daughter is accountable and does not harm other girls in the school. The characters and storylines were developed using desired, positive norms identified within the community during formative research to ensure that the drama was relatable and appropriate to the setting [43].

As part of the exploratory evaluation, participants were provided with a USB flash drive of the radio drama in September 2021 and instructed to listen to one episode per week at a time and place convenient for them. Providing participants with the flash drive also enabled participants to pause and replay the episodes at their leisure. Trained facilitators then visited a sub-group of randomly selected participants' homes to deliver household discussion sessions once a week after the family had listened to the episodes. The aim of this in the overall impact evaluation was to determine whether discussion sessions had an additional impact beyond that of the radio drama alone, with the researchers concluding that there was no clear evidence that they had an additional impact [44]. In these sessions, facilitators used a discussion guide to ask one question related to each episode that drew out key storylines and themes from the drama for the family to critically reflect on together. The discussions focused on issues related to challenges faced by adolescent girls, power dynamics, assertive communication, and how caregivers and the wider community can help to prevent age-disparate transactional sex and support others to change harmful behaviours.

## Sampling and data collection

Participants were selected from households who were recipients of services provided by TCRS in Kishapu district, who implemented this project. Prior to this study, solar powered radios had been distributed to 331 low-income households, which all included a person living with a disability, with the aim of sharing health-related information to the most vulnerable community members. Households were then sampled from this group for the LINEA study as they already possessed the radios needed to listen to the drama. We purposively sampled households in which at least one adolescent girl aged between 12–16 years was living, so that caregivers of adolescent girls in the age range targeted by the LINEA radio drama were included. Sixty household were randomly sampled to take part in household discussion sessions.

**Table 1. Description of male role models from the *Msichana wa Kati* radio drama who help change harmful behaviours related to age-disparate transactional sex.**

| Character | Description of storyline |
|---|---|
| Tuma | A 23-year-old *bodaboda* (motorcycle taxi) driver who initially faces pressure from other *bodaboda* drivers to engage in age-disparate transactional sex with Amali, the central adolescent girl character. Over the course of the story, he learns responsibility and respect, especially for women. He ultimately realizes that having a relationship with Amali is not in her best interest and could also negatively impact his own life plans with his same-age girlfriend. |
| Hami | An older, more experienced *bodaboda* driver who advises Tuma to avoid age-disparate transactional sex relationships. Hami had previously gotten an adolescent girl pregnant and had to pay for his mistake. |
| Tizo | Amali's father, who begins the story as a truck driver who abides by traditional, patriarchal gender norms. An accident at work, however, leaves him jobless and dependent on his wife. Over the course of the story, he learns that being a family man means a lot more than providing financially, it also means being involved in his daughters' emotional and non-material needs. |
| Zaki | A teacher at Amali's school who takes his job seriously and is respected by his pupils. When he discovers that another teacher at the school is engaging in age-disparate transactional sex with adolescent girl students he intervenes. |

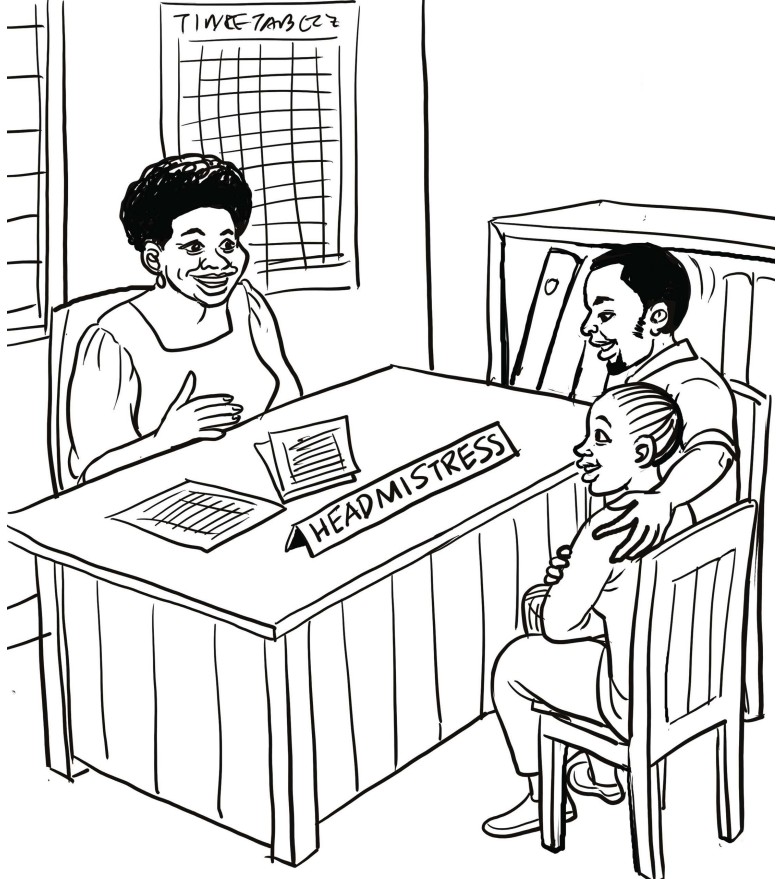

**Fig 1. Illustration from the *Msichana wa Kati* radio drama of Amali's father Tizo attending a meeting with the school headmistress.**

Caregivers were defined as anybody that cared for an adolescent girl, and subsequently included parents, grandparents and older siblings. Men caregivers were chosen to examine the impact of the intervention on caring roles, not the impact of the intervention on men engaging in age-disparate transactional sex. Therefore, it was not expected that any of the male participants would have ever engaged in age-disparate transactional sex. Adolescent girls, men and women were then randomly sampled from the selected households to participate in IDIs, with only one participant chosen from each household to maintain confidentiality. Participants with a cognitive disability were excluded. Participants were recruited from 21–27 July 2021, but due to delays data collection did not start until September. In total, 81 adolescent girls and men and women caregivers were selected to participate in baseline and endline IDIs, including the 18 men caregivers whose IDIs were analysed for this study. Of the 18 men included in the current study, 13 were assigned to take part in household discussion sessions.

The IDIs were carried out by sex-matched researchers from AGO who participated in the intervention development and were familiar with the issues underpinning the study, including adolescent sexual health and age-disparate transactional sex. The interviews were conducted by the same researchers at baseline and endline using similar topic guides. At both timepoints the IDI topic guides covered the themes of gender, power and age-disparate transactional sex, including participant's attitudes, beliefs, social norms and experiences. At endline topic guides also covered experiences of listening to the radio drama as a family, and of facilitated household discussion sessions. Throughout the interviews

probing was used to glean examples of participant's personal experiences with the topics covered (see S1 and S2 File for the IDI topic guide themes and example questions). All interviews were conducted in Kiswahili and then transcribed verbatim and translated into English. The interviews were conducted in a time and place that was convenient and safe for participants.

### Ethical approvals

Ethical approval for this study was obtained from LSHTM (ref: 22863–1) and the National Institute for Medical Research (NIMR) (ref: NIMR/HQ/R.8a/Vol.IX/3698) in Tanzania. Prior to the start of the study, TCRS shared the NIMR research ethics approval certificate and the radio drama with the Kishapu District Executive Director to inform them of the study.

### Written informed consent from study participants

Interviewers received training in ethical research and informed consent procedures before data collection. Potential participants were given an information sheet in Kiswahili, which was then read with the researcher to enable them to ask any questions before consenting to participate in the study. A verbal comprehension questionnaire was then provided to ensure that the information was understood correctly. If the participant agreed to partake in the study, they were then presented with a consent form to be signed (or marked with a thumb print if they could not write) and dated. If the individual was unable to read or write, an impartial witness was also present during the consent process, printing and signing their name on behalf of the participant. For additional information on the study setting, sampling and data collection procedures, intervention implementation and ethical considerations see Pichon et al. 2022 [44].

### Inclusivity in global research

Additional information regarding the ethical, cultural, and scientific considerations specific to inclusivity in global research is included in the Supporting Information (S1 Checklist).

### Data analysis

The transcripts were first uploaded into NVivo 12, where the first and second authors (AS and MP) read through the interviews to become familiar with the data, noting any initial impressions and highlighting sections of the transcripts that were particularly rich, based on knowledge of the literature. The coding framework was then drafted based on social norms theory, the topic guides and existing literature on age-disparate transactional sex. AS and MP dual-coded all of the transcripts, and LHM dual-coded 20%. They then met with the Principal Investigator (AMB) to discuss emerging themes and refine the coding framework. Using this coding framework, the transcripts were analysed using a thematic approach [55]. The coding framework was also refined throughout the coding process as themes arose inductively, by adding and removing codes. The combination of inductive and deductive approaches allowed the analysis to be guided by theory, whilst also allowing for new and diverse perspectives to arise, which was important in this study as men are understudied within this field and new themes were likely to emerge.

The data was then charted onto a matrix that summarised each transcript using the framework method [56]. The framework method was chosen because it created ease in comparing segments of data, whilst retaining the context of each account [56]. Each row of the matrix represented one transcript, with the participant's baseline interview on top and the endline interview just below it. The columns of the matrix each contained a key theme, for example "collaborative parenting with mothers". Within each cell of the matrix was a summary of the attitudes, beliefs, norms and behaviours surrounding that theme from the interview, including key quotes where present. This allowed for a comparison of individual participants' views before and after the intervention and an analysis of changes by theme. It also allowed individual differences of opinion to be highlighted with ease, as well as any deviant cases.

The combination of thematic coding and the framework method allowed for an in-depth exploration of themes within the data, whilst facilitating a comparison between the IDIs before and after the intervention. While we use a trajectory analysis to describe the change between baseline and endline for a few individuals, our analyses relied primarily on the frequency with which different thematic findings at the aggregate level arose between the two time points and if, at endline participant's referred to the radio drama characters or storylines as the reason for their attitude or belief, highlighting that the changes were due to the intervention [57]. Quotes from all but two of the participants are presented in the results section below, with these two excluded as they did not provide any novel contributions that were not already presented by other participants. Therefore, this paper presents a representative picture of the variety of perspectives contained within the sample.

## Results

This study includes 18 participants, all of whom participated in two 90-minute interviews. Participants were men aged 25–76 years and came from 13 villages in Kishapu district. Most caregivers were aged between 40 and 70 years and were fathers of an adolescent girl. Farming was the most common occupation; however, three participants were unable to work due to a physical disability or because of caring responsibilities for a family member living with a disability (Table 2).

We begin this section by presenting evidence on engagement with and acceptability of the radio drama, followed by participants' conceptualisations of positive and negative masculinities, and how these influenced their interpretations of the drama. The indications of change surrounding father-daughter communication about sex and age-disparate transactional sex are then explored, as well as participants' beliefs about their role in preventing age-disparate transactional sex as fathers or caregivers. Finally, we discuss the intervention's impact on participants' beliefs concerning the need to educate other men and the broader community to prevent age-disparate transactional sex, rather than focusing exclusively on girls.

### Engagement with and acceptability of the radio drama

Overall, the vast majority of the men found the drama to be acceptable and appropriate for their adolescent girls to listen to. All reported listening to the radio drama, and all but one reported listening with their daughter(s), which they reported was a *"comfortable"* experience where they *"felt free"* to discuss the storylines whilst listening, reflecting the appropriateness of the messaging. Frequently, participants noted the utility of the drama as a tool to educate their children. For example, a grandfather explained:

**Table 2. Characteristics of male caregivers who participated in the study.**

| Characteristic | Participant count |
|---|---|
| **Age group** | |
| 20-29 years | 1 |
| 30-39 years | 1 |
| 40-49 years | 4 |
| 50-59 years | 6 |
| 60-69 years | 4 |
| 70+ years | 2 |
| **Occupation** | |
| Seasonal work | 3 |
| Farmer | 12 |
| Unable to work | 3 |

*I always advise[d] them [his granddaughters] to listen more to the radio drama, because [the radio drama characters] are the one[s] who educate us. (76-year-old man, Endline, Household discussion sessions, IDI 48)*

The one participant who did not listen to the drama with his granddaughter believed she was too young to listen, and so he listened alone:

*I felt that, for this girl, listening to this radio drama is not proper. I realised that I couldn't involve her because she is still young. She is still underage; her mind is still immature... She is 12 years old. (62-year-old man, Endline, Household discussion sessions, IDI 60)*

Whilst all other participants felt the drama was appropriate for their daughters to listen to, this example highlights that some caregivers may believe age-disparate transactional sex and sexual relationships in general are inappropriate topics to discuss with young adolescent girls.

## Aspirational masculinities and interpretations of the radio drama

Throughout the interviews men frequently discussed qualities that make a 'good' man or father, describing positive male role models in the community and certain family-oriented behaviours they had observed and admired about other men. This extended to some of the male characters in the radio drama, reflecting the inclusion of positive role models who are relatable to listeners. When asked about men in their community that they admire, participants mostly described business-men or affluent agriculturalists, citing their long-term personal and financial success:

*I admire him because he has built for himself a house with galvanized iron roofing, then he has a cupboard and television. (58-year-old man, Baseline, Household discussion sessions, IDI 54)*

*He is a very good farmer, I admire him, and my goal is to reach his level. (53-year-old man, Baseline, Household discussion sessions, IDI 55)*

A few also described men who provided financially for their families, thereby showing them care, and admired the harmony and love they witnessed in their day-to-day lives:

*[I admire him] because of the way he is with his family. The love he has for his family. (42-year-old man, Baseline, No household discussion sessions, IDI 79)*

Therefore, long-term personal development and caring for one's family were seen as things that 'good' men do, highlighting attributes of aspirational masculinities.

In contrast, it was often emphasised – at both baseline and endline – how men who engage in age-disparate transactional sex experience a 'double loss', the financial loss of giving money and gifts to girls, and the consequent loss of the ability to provide for his family:

*If for example, he has a family, his children will not have money for food, or breakfast. This is a loss to him. (25-year-old man, Baseline, Household discussion sessions, IDI 43)*

*For instance, maybe he's gotten wealthy as a result of selling cotton, but he spends the money on that girl, so his family does not benefit from the wealth. (58-year-old man, Baseline, Household discussion sessions, IDI 54)*

Furthermore, as engaging in sex with underage girls is illegal in Tanzania, participants explained how men may abandon their family to escape law enforcement after being caught engaging in age-disparate transactional sex:

*[If you have sex with an underaged girl] you have to be arrested and you may run [from the police], the effect is you abandon your family. (63-year-old man, Endline, Household discussion sessions, IDI 59)*

Consequently, participants frequently labelled men who engage in age-disparate transactional sex as *"insane"* or *"childish".* Despite this, the majority described men's sexual desire as being pervasive, and a natural characteristic of being a man, thus explaining why they still engage in age-disparate transactional sex.

After listening to the drama many participants reported admiration for a character named Tuma who had previously engaged in age-disparate transactional sex, but who later changed his ways and began educating other men to not engage:

*I liked him [Tuma] because... at first, he was living with bad behaviours [engaging in age-disparate transactional sex]. Later [he] changed, you see?... He asked himself for how long he was going to live that kind of life. Then he decided to stay and be serious with work. (42-year-old man, Endline, Household discussion sessions, IDI 49)*

*I liked him [Tuma] because at first, he was ignorant... but eventually he changed... I liked his ability to learn, he was even [teaching] his fellows and [educated] other people. (54-year-old man, Endline, Household discussion sessions, IDI 58)*

These quotes demonstrate that participants believed it was possible for men to change even if they previously engaged in behaviours that went against what a 'good' man does.

### Changes in the role of men caregivers in preventing age-disparate transactional sex after listening to the radio drama

We begin this section by reporting evidence of the drama catalysing discussions about sex and age-disparate transactional sex between men caregivers and their adolescent daughters. We then discuss how this increased men's understanding of the challenges faced by adolescent girls, and how their perceptions about fatherhood responsibilities were altered to include engaging with their daughters to prevent age-disparate transactional sex.

**Catalysing discussions about sex and age-disparate transactional sex.** We found strong evidence that the radio drama provided opportunities for men to discuss age-disparate transactional sex and sex more broadly with girls, where previously they did not. Before the intervention men's communication with daughters, younger sisters and granddaughters about sex was often described vaguely and consisted primarily of rebuking them if they observed *"bad behaviour"* (such as speaking with male strangers) or being *"strict"* (such as enforcing early curfews). Furthermore, discussions with girls about sex were frequently described as the responsibility of female caregivers – most often aunties. None of the participants mentioned their daughters or granddaughters initiating conversations about sex and relationships, let alone age-disparate transactional sex.

Through listening to the drama as a family, discussions around the topic were made possible. For example, when asked whether he discussed age-disparate transactional sex with his daughter at baseline one participant said:

*[…] on the part of the girl, she gets most of the lessons through her mother. Now if something goes wrong on the part of the daughter the father of the family will blame the mother. (57-year-old man, Baseline, No household discussion sessions, IDI 76)*

While after listening to the radio drama at endline the same participant reported initiating conversations about it with his daughters, highlighting that he had taken on a greater role in educating his daughter about age-disparate transactional sex:

*Many times, we were listening to the radio drama... and I could tell people in our group to stop playing the radio drama and I'd ask them [his daughters] 'which character [do you think] has given good advice?' (57-year-old man, Endline, No household discussion sessions, IDI 76)*

Similarly, when asked if the drama episodes were educative, another man responded:

*Yes, they have teachings... [whilst listening] I am also helping her [his daughter] to ask questions... I ask her 'how are they [the characters] not right?' (59-year-old man, Endline, No household discussion sessions, IDI 74)*

Consequently, the drama opened a line of communication about age-disparate transactional sex and sex more broadly between some of the participants and their daughters and granddaughters. One participant further explained that:

*The things that happened to [the characters in the radio drama] brought a lesson to my children, since they came across the same temptations on their way to school. (76-year-old man, Endline, Household discussion sessions, IDI 48)*

This led to girls discussing their experiences of *"temptation"* to accept gifts from older men with their male caregivers when they previously had not done so. It was evident that these men did have the knowledge and capability to discuss age-disparate transactional sex with girls after listening to the drama.

While the participants report initiating these discussions on their own while listening to the radio drama, it is also possible that the facilitation of these conversations was role modelled to them during the household discussion sessions. However, as seen in the quotes above, we found evidence of the radio drama catalysing conversations even among men who were not assigned to take part in household discussion sessions.

**Shifting perceptions of fatherhood and men's role in preventing age-disparate transactional sex.** For men, the radio drama contextualised how young girls who engage in age-disparate transactional sex are, and the challenges they face in their everyday lives. Before the intervention, many explained that only mothers understood the challenges facing girls because a mother *"knows how a woman can be seduced as she was once a girl herself"* (76-year-old man, Baseline, Household discussion sessions, IDI 48).

The characters and storylines in the drama reportedly filled this knowledge gap – for example, scenes where girl characters were approached by older men on their way to school modelled how age-disparate transactional sex occurred in real life. From this, several participants explained that the drama made them consider their own daughters, sisters and granddaughters as potential victims of age-disparate transactional sex, rather than it only affecting other, *"bad-mannered"* girls who were distant from them:

*[If I engaged in age-disparate transactional sex] it will be like... I have [had sex] with... my granddaughter. (63-year-old man, Endline, Household discussion sessions, IDI 59)*

*And I have a daughter, she will be treated [badly by men] like this. (53-year-old man, Endline, Household discussion sessions, IDI 55)*

Thus, the drama made adolescent girls more relatable for the participants, leading to more empathetic responses and a more nuanced understanding of why some engage in age-disparate transactional sex. This also made the issue more relevant to their own lives; they began to think of it as something that could affect their own children and therefore something they must take steps to prevent.

Subsequently, communicating with girls about age-disparate transactional sex and sex more broadly was incorporated into aspirational conceptualisations of how 'good' men provide for their family. Before the intervention, participant

accounts frequently described a gendered division of parenting, with mothers providing emotionally and fathers financially, and when asked how men could prevent adolescent girls from engaging in age-disparate transactional sex, many men described their role predominantly as financial providers:

> *If a person like me was rich, I have a salary, my child will have her every need met. (62-year-old man, Baseline, Household discussion sessions, IDI 53)*

> *Men can help a girl [to not engage in age-disparate transactional sex] by fulfilling her needs, like buying her clothes and shoes. (62-year-old man, Baseline, Household discussion sessions, IDI 60)*

Therefore, they believed they could help to prevent age-disparate transactional sex only by providing for girls' economic and material needs, so that they are not "tempted" by men through gifts.

After the intervention, the importance of discussing sex and relationships with girls was incorporated into ideas of providing for one's family. For example, when one man was asked whether after listening to the drama, he discussed age-disparate transactional sex with this family, he responded:

> *Yes, [for] the wellbeing of the family. (59-year-old man, Endline, Household discussion sessions, IDI 51)*

Participants were also more likely after listening to the drama to emphasize the joint responsibility of parents to engage with their daughters to prevent age-disparate transactional sex, rather than encouraging the same-sex approach they previously held:

> *I must be together with my wife... we can [both] help the girls. (30+-year-old man, Endline, Household discussion sessions, IDI 50)*

Similarly, a man who had previously reported his role was to provide financially said:

> *A father and mother should have a conversation [about age-disparate transactional sex] with their children [together]. (48-year-old man, Endline, Household discussion sessions, IDI 53)*

After listening to the drama, most men also reported that not only was it part of their parental role to discuss age-disparate transactional sex with their daughters, but also that wider parental engagement in the lives of their daughters was an important part of preventing age-disparate transactional sex. Several called upon an example from the drama where Amali's father, Tizo, visited his daughter's school to speak with her teachers after her older sister became pregnant:

> *I liked [this scene] because he [Tizo] volunteered personally to go and ask there at the school... as a father it's your responsibility to know [what is happening to your daughter]. (68-year-old man, Endline, Household discussion sessions, IDI 45)*

This example underlined men's belief about the importance of their involvement and participation in their daughters' lives after listening to the radio drama.

### Educating other men and the wider community to prevent age-disparate transactional sex after listening to the radio drama

In this section we begin by describing the shift in men's beliefs that girls must be educated to prevent age-disparate transactional sex, to highlighting the importance of also educating men after listening to the radio drama. We then present

evidence of men's recognition of the importance of strategies to educate the whole community to protect adolescent girls from age-disparate transactional sex after listening to the radio drama.

**Shift from educating girls to educating men.** Before listening to the drama participants argued that girls should be educated against engaging in age-disparate transactional sex. Here, the impetus was placed on girls to not engage and to remove them from any influence, rather than on men to not *"lure"* or *"seduce"*:

*Education should be given to girls of that age [13-15 years old] in the community so as to understand that when you have sex at that age first, your dreams will not be achieved. (57-year-old man, Baseline, No household discussion sessions, IDI 76)*

Teachers were also often seen as an important tool for monitoring and communicating with girls:

*I think teachers can do a good job [discussing age-disparate transactional sex with girls] because they have them for most of the day. (42-year-old man, Baseline, Household discussion sessions, IDI 49)*

At endline others disagreed with this role for teachers, as they had heard in the drama and through gossip that teachers sometimes engage in sex with pupils in exchange for marks, and therefore can act as perpetrators of age-disparate transactional sex themselves.

Many believed that the solution to preventing the practice was to build schools closer to where pupils lived, or provide dormitories, transport and security to prevent any potential encounters with older men that could lead to age-disparate transactional sex:

*The government and private companies should bring [school] transport or build dormitories [for students]... it will be better [for girls] because the security will be strong. (46-year-old man, Baseline, Household discussion sessions, IDI 47)*

Notably, their narratives centred around keeping adolescent girls away from the "*temptations*" of older men. There was no mention of how the older men themselves could be engaged with to prevent age-disparate transactional sex, therefore, placing the responsibility and blame onto the girls rather than the men who seek to have relationships with them.

We did find some evidence at endline, however, of shifted perceptions for who is to blame for age-disparate transactional sex from girls to men. For example, one man at baseline described these relationships as starting when:

[…] *she has no money to buy sugarcane or something like that, so maybe a man gives her sugarcane as a gift. Because she really wanted to have some sugarcane, she will agree and accept his gift. (59-year-old man, Baseline, Household discussion sessions, IDI 51).*

At endline, the same participant provided a much more nuanced description of the inequitable power dynamics inherent in age-disparate transactional sex relationships:*[…] that gift he is providing it's not big, he sets a trap […] he starts that way, buying soda, in the end he will win. Because she has a childish understanding […] she doesn't know what will happen in the future. (59-year-old man, Endline, Household discussion sessions, IDI 51)*

Underpinning many participants' accounts on this issue was the belief that male sexual desire is inherent and pervasive, and something that girls must be educated against rather than men being able to change. But after the intervention, the importance of peer educating other men was discussed by several participants. Many reported that their favourite character was Tuma, a young man who stopped engaging in age-disparate transactional sex and began to educate his peers on the harms of the practice, using his own experience to advise other men and consequently shifting the norm of pervasive sexual desire:

*I liked [Tuma] because at the beginning he was acting against the accepted behaviours [engaging in age-disparate transactional sex], but he later on changed to be a good person. (46-year-old man, Endline, Household discussion sessions, IDI 47)*

*I liked him [Tuma] most because... he is educating [other men] about those things [age-disparate transactional sex]. (62-year-old man, Endline, Household discussion sessions, IDI 60)*

Others admired another character, Zaki, who was a male teacher that witnessed his colleague engaging in sex with a pupil in exchange for marks and reported him to the school authority.

Where previously age-disparate transactional sex had been seen as an issue for girls to be educated against and protected from, the drama provided male role models who demonstrated to participants how to re-consider their own roles – such as discussing the topic with other men. A few participants also explained that they had listened to the drama with other men outside of the study sample, leading to discussions about age-disparate transactional sex, the challenges facing girls and how men can better engage with these topics. For example, when asked if he discussed the drama with someone outside of his family a man replied:

*Yes, we discussed [that we should] teach them [their daughters] to stop having sex... He [his friend] said 'we will teach our [own] children'. (53-year-old man, Endline, Household discussion sessions, IDI 55)*

Others reported that they would have discussed the drama with other men, but they incorrectly believed the intervention to be confidential, and therefore did not. This represents an important change in how men viewed their positionality regarding the prevention of age-disparate transactional sex, shifting the impetus away from girls towards a more collaborative approach in which men advise each other.

**Educating the community.**  Community engagement with the topic of age-disparate transactional sex was another strong theme raised by participants both before and after the intervention. Community gatherings and events were often described as a mechanism to ensure both girls and men behaved appropriately, often referencing *"rebuking"* and *"imprisonment"*, respectively, as ways for the community to prevent age-disparate transactional sex. One participant emphasised that community meetings to educate and discourage against age-disparate transactional sex should involve everyone within the community:

*Meetings should be non-discriminatory, even girls should attend. (68-year-old man, Baseline, Household discussion sessions, IDI 45)*

Before the intervention, ideas about how the community should engage with age-disparate transactional sex varied. Some believed that open meetings should be held to discuss the harms of engaging in the practice, while others argued that the community's main role was in enforcing the law and reporting perpetrators.

After listening to the drama, participants focused more on community-wide education – such as seminars delivered by non-governmental organisations and the government. More participants also referenced the need to educate men against age-disparate transactional sex, rather than just community wide messaging about the harms of the practice. One participant explained that the community was responsible for telling men *"you should not have sex with our daughters"* (53-year-old man, Endline, Household discussion sessions, IDI 55), establishing how parents must engage within the community to protect their daughters as part of their parental duties. Thus, the drama encouraged participants to see age-disparate transactional sex as a societal issue requiring societal solutions, rather than an individualised issue.

Both before and after listening to the drama, some participants described a traditional local proverb: *"Mtoto wa mwenzio ni wako"* (translated from Kiswahili as *"the child of your fellow is your child"*). This emphasised that traditionally, all the

children within the community were to be treated as your own child. However, sentiments counter to this were expressed by some participants before the intervention:

> *If you tell her to stop doing that because she is still young, she'll tell you to not follow her, mind your own business. She'll remind you that you are not her parent. So, I decided to just watch [rather than educate her]. (42-year-old man, Baseline, No household discussion sessions, IDI 79)*

> *Everyone has their own household, I can`t understand everyone and the behaviour of his home" (59-year-old man, Baseline, No household discussion sessions, IDI 74)*

These quotes reflect how, despite a community-oriented norm of protecting other people's children as your own, an individualistic approach to parenting was common – centred around the household rather than the community. The radio drama reinforced the importance of this proverb to several participants when one character, a middle-aged woman, intervened and aided a couple of adolescent girls who were not her daughters, but who were being targeted by men for age-disparate transactional sex. Several participants described admiration for her because she was selflessly helping children within the community. Whilst the proverb already existed within this context, this character provided an example of how to follow it in practice, reinforcing this protective norm and the communities role in age-disparate transactional sex prevention.

## Discussion

This study has examined the indications of change surrounding men caregivers' attitudes, beliefs, social norms and behaviours related to their role in preventing age-disparate transactional sex after listening to the LINEA radio drama and taking part in household discussion sessions about it. The study yielded insights into programming developed for, and delivered to men, and found promising results, particularly in improving communication between male carers and adolescent girls and altering men's beliefs about their ability to prevent age-disparate transactional sex. Participants also shared their increased motivation to partake in community engagement around the topic.

Whilst men's engagement with parenting interventions can at times be limited [18,21–24,58], this study found men's engagement with the LINEA radio drama to be high throughout the study period. The LINEA radio drama delivery via USB flash drive, designed to be accessible and convenient, was key to this success. This is in line with previous research on engaging men in VAWG programming, which highlights the importance of providing men with flexible programming they can engage with at a place and time that is convenient for them [26]. Other strategies that have been identified as key to promoting men's engagement include strong partnerships with implementing partners and in-country stakeholders, creating contextually appropriate content based on formative research, taking the time to adequately train facilitators, engaging influential community members to support normative change and securing funding from flexible donors [59]. These strategies were also adopted by LINEA and are further described in an intervention development paper by Howard-Merrill et al. [43].

Although this small qualitative study did not find direct evidence of additional impact from structured household discussion sessions, they may have helped facilitate conversations that might not have otherwise occurred, and provided caregivers with a model for initiating and engaging in such discussions. Communication theory suggests that discussions about programming serve as an important, indirect pathway to behaviour change as they indicate that others within one's social group are engaging with the intervention messages [60]. However, a recent review of edutainment programmes aimed at preventing violence against women and children found mixed evidence on whether discussion sessions contribute additional impact beyond the edutainment intervention itself [61]. Some researchers have hypothesized that this may be because discussions occur organically within communities, particularly when interventions are delivered in group

settings, reducing the need for structured sessions facilitated by implementers [62]. Given these mixed findings, further research is needed to assess the true impact of discussion sessions, particularly in comparing individual and group-based delivery modalities. Given the myriad factors that constrain men's participation in parenting programmes, the effectiveness of the home-based USB flash drive mode of delivery is also an important contribution to the evidence base underpinning men's engagement.

Evidence shows that increased father engagement and connectedness can reduce adolescent sexual risk-taking, making these finding particularly encouraging [13,14]. However, it is important to note that father-daughter communication about sex is an understudied area [63], and more research on this issue in different contexts is required to better understand the potential of such discussions in improving sexual health outcomes, and the mechanisms and pathways to impact.

Previous work has shown that engaging in age-disparate transactional sex is one way for men to fulfil the gendered norm of material provision, demonstrating their masculinity by providing money and gifts to young girls [34,38,41,42]. The importance of men's material provision was also reiterated by participants in this study, albeit in relation to providing for one's family rather than in sexual relationships, due to this sample consisting of men caregivers specifically. The intervention was able to re-direct this pervasive norm towards a more holistic caregiving role, encouraging men to actively engage in discussions with girls about age-disparate transactional sex and sex more broadly, and being actively engaged in their lives, where previously this had been the purview of female caregivers. This shift highlights the need for gender-transformative programming that not only seeks to shift away from negative masculinities but also acknowledges existing positive masculinities and works to expand and strengthen them.

Similarly, the intervention reinforced the existing Kiswahili proverb *"Mtoto wa mwenzio ni wako"* or in English, *"the child of your fellow is your child"*, in the context of preventing age-disparate transactional sex. This reflects a shift in beliefs around community responsibility and highlights the effectiveness of building on existing norms rather than introducing new ones. Such existing norms can be protective [64] and are easier for programmes to promote and build upon rather than try to introduce new norms. Similar success was observed in Uganda's 'Parenting for Respectability' programme, which aligned with fathers' desires for well-mannered and respectful children [23].

Men's willingness to mobilise other men and advocate for gender equality is often limited in practice, despite supporting the concept in theory [6,7]. However, in our study participants described the importance of educating other men, as well as participating in community-wide education rather than solely parenting girls within their own household, which are essential elements required for social norms change. By role-modelling how to peer-educate other men and engage with the wider community on age-disparate transactional sex, the radio drama enabled participants to widen their expectations of what 'good' men do to include advocating for change and peer-educating other men in the community. As such, the intervention encouraged men to form part of the wider effort to prevent age-disparate transactional sex from occurring, rather than treating them as merely perpetrators of these harmful behaviours.

This emerging finding of peer-educating other men is particularly promising, as social pressure has been shown to limit the effectiveness of interventions in changing men's behaviour in relation to complex, harmful practices such as VAWG [1,4,23]. As such, it is important for interventions to consider having built-in mechanisms to encourage wider social and community change. However, as only a few participants did discuss the topic with other men, the wider LINEA trial and other future interventions should consider providing more guidance and skills training on how to discuss the topic with others, beyond simply modelling such discussions within a storyline. Such guidance should also consider concerns from the activism sphere around the marginalisation of women and girls' voices in VAWG prevention work that seek to engage men [1,5]. Whilst engaging men outside of the direct intervention is important, men in the community should still be held accountable to the needs and priorities of women and girls when addressing VAWG. In supporting this change, practitioners must be careful not to inadvertently promote men's controlling behaviours by ensuring they are respectful of the agency of adolescent girls', including, for example those who do choose to date.

## Limitations and strengths

The aim of this paper was to gain a nuanced perspective of men's experiences of the LINEA radio drama and household discussion sessions. Whilst the sample size was relatively small, we are confident the pre/post design and the richness of the data provided us with useful insights into the indications of change following the intervention. The sample largely consisted of older men, and it is possible that the caregiving aspects of this intervention may not be as resonant with younger men, however, the drama was designed to be relevant to all ages and thus includes other characters and themes which younger men may better connect with. While data from adolescent girls and women caregivers were included in our companion papers (see [44,45]), findings did not speak directly to their perspectives of the role of male caregivers in preventing age-disparate transactional sex, and therefore data from girls and women were not included in this study.

There is always a possibility that participants provide socially desirable responses, particularly when discussing sensitive topics such as age-disparate transactional sex. We acknowledge this potential bias, although we found no direct evidence of it. This may be, in part, due to participants long-standing relationship with TCRS, which fostered trust in both the organisation and the study. While the AGO researchers were more educated and came from a more urban area then the participants, being introduced by TCRS and having plenty of experience working with the community likely helped establish credibility and comfort. Although it is possible that participants underreported their engagement in age-disparate transactional sex, this would not have affected the primary findings of this study, which focus on broader changes in male caregivers following the intervention. However, we recognise that social desirability bias may have led some participants to overstate the programmes effects.

It is also important to acknowledge that households for this study were all low-income and sampled from one district of the Shinyanga region, meaning these findings may not be transferrable to other populations in Shinyanga or Tanzania. However, the aim of this qualitative study was to gain insight into the potential of the LINEA radio drama to shift harmful masculinities and promote the perception of men caregivers as important parenting allies. The findings described here are very promising, and a more robust evaluation of LINEA is underway.

## Conclusions

This study explored indications of change in men caregivers' attitudes, beliefs, social norms and behaviours following the LINEA radio drama and household discussion sessions aimed at preventing age-disparate transactional sex in Shinyanga region, Tanzania. Following the intervention, attitudes, beliefs and behaviours of men caregivers shifted, highlighting the potential for edutainment interventions to create relatable, realistic stories and characters that can model behaviours to address complex issues such as age-disparate transactional sex and its associated norms and challenges. Our findings suggest that by building upon and expanding existing protective norms within the community, men were able to reconceptualise their own role in preventing age-disparate transactional sex. The intervention also allowed them to incorporate engagement with girls – discussing age-disparate transactional sex with them – and peer-educating other men into their understanding of being 'good' providers. The local proverb "*the child of your fellow is your child*" was used in a novel context, emphasizing community responsibility in protecting girls from age-disparate transactional sex. Finally, our study highlights the need for future programming to include mechanisms to encourage broader social and community change, while ensuring that men remain accountable to the needs and priorities of women and girls in VAWG prevention work.

## Licencing

## Supporting information

**S1 Checklist. Inclusivity in global research questionnaire.**
(DOCX)

**S1 File. Baseline IDI topic guide themes.**
(PDF)

**S2 File. Endline IDI topic guide themes.**
(PDF)

## Acknowledgments

We would like to thank the men who gave their time and energy to participate in this study. We would also like to thank the teams from Amani Girls Organization and the Tanganyika Christian Refugee Service who implemented the LINEA intervention and carried out the in-depth interviews analysed here.

## Author contributions

**Conceptualization:** Alicia Sharif, Marjorie Pichon, Ana Maria Buller, Lottie Howard-Merrill.

**Data curation:** Alicia Sharif, Marjorie Pichon.

**Formal analysis:** Alicia Sharif, Marjorie Pichon, Ana Maria Buller, Lottie Howard-Merrill.

**Funding acquisition:** Ana Maria Buller, Lottie Howard-Merrill.

**Investigation:** Veronicah Gimunta, Oscar Rutenge, Revocatus Sono.

**Methodology:** Alicia Sharif, Marjorie Pichon, Ana Maria Buller, Lottie Howard-Merrill.

**Project administration:** Marjorie Pichon, Veronicah Gimunta, Oscar Rutenge, Revocatus Sono, Ana Maria Buller, Lottie Howard-Merrill.

**Resources:** Veronicah Gimunta, Oscar Rutenge, Revocatus Sono.

**Supervision:** Marjorie Pichon, Veronicah Gimunta, Oscar Rutenge, Revocatus Sono, Ana Maria Buller, Lottie Howard-Merrill.

**Validation:** Marjorie Pichon, Veronicah Gimunta, Oscar Rutenge, Revocatus Sono, Ana Maria Buller, Lottie Howard-Merrill.

**Writing – original draft:** Alicia Sharif, Marjorie Pichon, Ana Maria Buller, Lottie Howard-Merrill.

**Writing – review & editing:** Alicia Sharif, Marjorie Pichon, Veronicah Gimunta, Oscar Rutenge, Revocatus Sono, Ana Maria Buller, Lottie Howard-Merrill.

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
