## [Editor Report · Decision Letter 0]

18 Sep 2024

PONE-D-24-32227“The child of your fellow is your child”: Building on existing protective norms to engage men as caregivers; qualitative findings from an exploratory evaluation of an edutainment intervention to prevent age-disparate transactional sexPLOS ONE

Dear Dr. Pichon,

Thank you for submitting your manuscript to PLOS ONE. After careful consideration, we feel that it has merit but does not fully meet PLOS ONE’s publication criteria as it currently stands. Therefore, we invite you to submit a revised version of the manuscript that addresses the points raised during the review process.

We look forward to receiving your revised manuscript.

Kind regards,

Daniel Romer

Academic Editor

PLOS ONE

**Additional Editor Comments:**

Your paper presents findings from in-depth interviews with a sample of 18 adult men who participated in a radio-drama intervention designed to reduce of the risks of transactional sex with adolescent girls in Tanzania. Before sending this out for review, I have some suggestions for revision that will make the review process go smoother.

My reading suggests that the study gave 331 low-income households the opportunity to listen to the drama on a weekly basis and met with the households to discuss the drama. For this study, 81 members of those households were invited to be interviewed at both the beginning and end of the intervention, including the 18 men who were the focus of this paper.

You provide examples of how the men felt about the role of men as potential change agents in the community to prevent transactional sex among adolescent girls with a focus on the post interviews. However, I have several suggestions for ways to make the results easier to follow and to interpret.

First, it is not clear whether the intervention was merely listening to the drama or of having discussions led by the research team on a weekly basis. Those discussions should be described if they were part of what the 18 men were exposed to. If those discussions focused on the themes of interest, then it is not the drama per se, but the drama as a way to provoke conversations with the researchers that is being studied. It would be preferable if those weekly discussions did not occur for the 18 men, so that one can draw conclusions about the drama as an intervention in itself. Otherwise, those discussions should be described as part of the intervention.

Another aspect of the study is the intervention itself. Can you provide some examples of how the drama featured the themes of interest? What story lines illustrated the need for men to take more responsibility for preventing transactional sex?

It would also be helpful to provide an outline of the interviews. What were the themes that guided the interviews? And what kind of probing was done to clarify the answers to those questions? It would also be helpful to know what the themes were at baseline versus at the end. This would be especially useful for interpreting whether the drama actually changed what the interviewees thought about men’s roles in preventing transactional sex. The results do not make a very clear case that the participants changed their views. It would help a lot to show examples of changes in particular participants views from baseline to the end. For example, did the men not even think about their roles at baseline but then show evidence of change at the end? Such evidence would be the most convincing that the intervention itself and not more extensive probing at the end was the source of change.

**Please resubmit the manuscript as we discussed.**

---

## [Author Response · Author response to Decision Letter 1]

25 Sep 2024

Please find our responses to the Editors comments in the attached "Response to Reviewers" document.

---

## [Decision Letter · Decision Letter 1]

4 Feb 2025

PONE-D-24-32227R1“The child of your fellow is your child”: Building on existing protective norms to engage men as caregivers; qualitative findings from an exploratory evaluation of an edutainment intervention to prevent age-disparate transactional sexPLOS ONE

Dear Dr. Pichon,

Thank you for submitting your manuscript to PLOS ONE.  After careful consideration, we feel that it has merit but does not fully meet PLOS ONE’s publication criteria as it currently stands. Therefore, we invite you to submit a revised version of the manuscript that addresses the points raised during the review process.

I should note that the delay in follow-up was largely due to reassignment of the manuscript from the original editor; apologies for this.  Two referees provided high-quality reports suggesting a range of generally minor revisions.  I will not add to these beyond noting that in referring to the broader literature around engaging men in the prevention of IPV, you could also draw on the recent systematic review I conducted with a team suggesting that engaging men was as effective in reducing and preventing IPV as other strategies (engaging women directly, community mobilization): Leight et al. in the Journal of Global Health, 2024.  Otherwise, the referees have provided useful suggestions to guide your revision.

We look forward to receiving your revised manuscript.

Kind regards,

Jessica Leight, PhD

Academic Editor

PLOS ONE

Journal Requirements:

Reviewers' comments:

Reviewer's Responses to Questions

**Comments to the Author**

1. If the authors have adequately addressed your comments raised in a previous round of review and you feel that this manuscript is now acceptable for publication, you may indicate that here to bypass the “Comments to the Author” section, enter your conflict of interest statement in the “Confidential to Editor” section, and submit your "Accept" recommendation.

Reviewer #1: (No Response)

Reviewer #2: All comments have been addressed

2. Is the manuscript technically sound, and do the data support the conclusions?

Reviewer #1: Yes

Reviewer #2: Yes

3. Has the statistical analysis been performed appropriately and rigorously? 

Reviewer #1: N/A

Reviewer #2: N/A

4. Have the authors made all data underlying the findings in their manuscript fully available?

Reviewer #1: Yes

Reviewer #2: Yes

5. Is the manuscript presented in an intelligible fashion and written in standard English?

Reviewer #1: Yes

Reviewer #2: Yes

6. Review Comments to the Author

Reviewer #1: This is a very well written paper that makes a strong and unique contribution to the literature. I certainly recommend this paper for publication. The authors clearly justify this paper given the gap in the evidence. I also appreciate the focus on engaging men to prevent adolescent girls' engagement in age disparate transactional relationships. I do have some suggestions to strengthen the paper overall:

The introduction notes that "most research on engaging men in VAWG efforts has focused on men as sexual partners." I would expand this to also reference the wealth of programme and research attention on engaging men as couples in IPV prevention programmes. Indeed, evidence suggests that working with men and women together to prevent IPV can be more effective than working with individual men and women (you could cite for instance Dunkle K, Stern E, Chatterji S, Heise L. Effective prevention of intimate partner violence through couples training: a randomised controlled trial of Indashyikirwa in Rwanda. BMJ Glob Health. 2020 Dec;5(12):e002439. doi: 10.1136/bmjgh-2020-002439. PMID: 33355268; PMCID: PMC7757483)

Another study to consider citing in the introduction around barriers/limitations of engaging men in fatherhood programmes is a study from Bolivia that did not have an impact on IPV, and a significant reason for this from the qualitative study was the poor engagement of men as fathers: Stern E, Alemann C, Delgado GAF, Vásquez AE. Lessons learned from implementing the parenting Program P in Bolivia to prevent family violence. Eval Program Plann. 2023 Apr;97:102207. doi: 10.1016/j.evalprogplan.2022.102207. Epub 2022 Dec 21. PMID: 3658743 2. Part of the reason for this was not doing good formative research or meeting men where they are; for instance men worked long hours and commuted so had limited time available for the time and location of the parenting programme sessions. I think this can come out too more in the recommendations of the importance of engaging men where they are and trying to address barriers to their engagement/meet their particular needs.

In the introduction, it would also be great if you could describe evidence of additional benefits of engaging men as fathers to prevent both violence against children and IPV? See for example the Bandheberho evaluation in Rwanda: .Doyle K, Levtov RG, Barker G, Bastian GG, Bingenheimer JB, Kazimbaya S, Nzabonimpa A, Pulerwitz J, Sayinzoga F, Sharma V, Shattuck D. Gender-transformative Bandebereho couples' intervention to promote male engagement in reproductive and maternal health and violence prevention in Rwanda: Findings from a randomized controlled trial. PLoS One. 2018 Apr 4;13(4):e0192756. doi: 10.1371/journal.pone.0192756. PMID: 29617375; PMCID: PMC5884496.

A recent evidence review published by the Prevention Collaborative (2025) also indicates the value of engaging men as fathers to prevent child maltreatment and IPV, as well as other benefits, including to strengthen men's mental health, and strengthen quality of relationships with their children. I think emphasizing this is important to not only consider men as instrumentalist to the benefits of women and children (absolutely important) but also consider the benefits to men through their engagement in such programming.

In the introduction the authors suggest that joint, or family-wide interventions might be more effective" clarify "than engaging fathers alone"; you could also link this to the benefits of accountability to women and girls through these family wide interventions.

On page 7, I would say 'most programmes have focused on early child outcomes' including to prevent violence against children, as many parenting programmes have violence against children as a primary outcome, and this links it back more broadly to focus on gender transformative programming and age-disparate sexual relationships.

Existing interventions targeting men as fathers of adolescent girls that I am more familiar with are also ones preventing VAC including preventing child marriage. For instance, there is an ongoing adaptation of the Indashyikirwa programme in Syria which has targeted couples who live with adolescent girls to prevent both IPV and child marriage of the parents' adolescent girls.

The method are very rigorous methods and I commend the very rich findings.

In the discussion, I am not sure it's fair to say while men's engagement with parenting programmes is often low, as we have many examples (including the Bandeberho from Rwanda), or this recent scale up of a parenting programme in Tanzania, of successfully engaging men in parenting programmes. I do think it is important to reference the latter study, especially since it is of parenting programmes with adolescent girls: Jamie Lachman, Joyce Wamoyi, Mackenzie Martin, Qing Han, Francisco Antonio Calderón Alfaro, Samwel Mgunga, Esther Nydetabura, Nyasha Manjengenja, Mwita Wambura, Yulia Shenderovich - Reducing family and school-based violence at scale: a large-scale pre–post study of a parenting programme delivered to families with adolescent girls in Tanzania: BMJ Global Health 2024;9:e015472.

I think you can definitely note many barriers to engaging men as fathers but also lots of examples of programmes that have done this. I would definitely cite Equimundo's recent report on what worked with Programming P to engage men as fathers here for examples of what has been learned about what works to engage men as fathers in parenting programmes.

Alemann, Clara, Rachel Mehaffey, and Kate Doyle. 2023. Core Elements of Gender-Transformative Fatherhood Programs to Promote Care Equality and Prevent Violence: A Practitioner Guide. Washington, DC: Equimundo. equimundo.org/resources/core-elements-of-gender-transformative-fatherhood-programs-to-promote-care-equality-and-prevent-violence/.

In limitations: could you add something about positionality of authors/researchers to the participants and how this was navigated? The authors' important point around accountability to women and girls is great; perhaps you can also note something around there can be limitations/risks with notion of men protecting girls including controlling behaviours or limiting agency on behalf of girls around who they are interested/in choose to date. It would be great if this is reflected on a bit more in discussion, including perhaps noting in the limitations that you do not have the perspective from adolescent girls themselves.

--

Reviewer #2: I have done the review and the paper is well presented, and a significant contribution to knowledge on the subject. Please see the attached minor comments for review.

7. PLOS authors have the option to publish the peer review history of their article (what does this mean? ). If published, this will include your full peer review and any attached files.

**Do you want your identity to be public for this peer review?** For information about this choice, including consent withdrawal, please see our Privacy Policy .

Reviewer #1: **Yes: ** Erin Stern

Reviewer #2: **Yes: ** Dr Geofrey Nimrod Sigalla MD, MPH, PhD

---

## [Author Response · Author response to Decision Letter 2]

25 Feb 2025

Comments Author Responses

Editor

Two referees provided high-quality reports suggesting a range of generally minor revisions. I will not add to these beyond noting that in referring to the broader literature around engaging men in the prevention of IPV, you could also draw on the recent systematic review I conducted with a team suggesting that engaging men was as effective in reducing and preventing IPV as other strategies (engaging women directly, community mobilization): Leight et al. in the Journal of Global Health, 2024. Otherwise, the referees have provided useful suggestions to guide your revision.

Thank you for recommending this relevant recent publication, we have added mention to it on pp 4-5, lines 95-97: “Moreover, a recent meta-analysis of 27 randomised controlled trials found that interventions were equally effective at reducing IPV when targeting communities, women only, men only or couples [20], suggesting the need for programming targeting each group.”

Reviewer 1

This is a very well written paper that makes a strong and unique contribution to the literature. I certainly recommend this paper for publication. The authors clearly justify this paper given the gap in the evidence. I also appreciate the focus on engaging men to prevent adolescent girls' engagement in age disparate transactional relationships. I do have some suggestions to strengthen the paper overall:

Thank you for your careful review and the positive comments, we look forward to addressing your suggestions.

The introduction notes that "most research on engaging men in VAWG efforts has focused on men as sexual partners." I would expand this to also reference the wealth of programme and research attention on engaging men as couples in IPV prevention programmes. Indeed, evidence suggests that working with men and women together to prevent IPV can be more effective than working with individual men and women (you could cite for instance Dunkle K, Stern E, Chatterji S, Heise L. Effective prevention of intimate partner violence through couples training: a randomised controlled trial of Indashyikirwa in Rwanda. BMJ Glob Health. 2020 Dec;5(12):e002439. doi: 10.1136/bmjgh-2020-002439. PMID: 33355268; PMCID: PMC7757483)

We have added the proposed sentence and citation on p 4, lines 79-80.

Another study to consider citing in the introduction around barriers/limitations of engaging men in fatherhood programmes is a study from Bolivia that did not have an impact on IPV, and a significant reason for this from the qualitative study was the poor engagement of men as fathers: Stern E, Alemann C, Delgado GAF, Vásquez AE. Lessons learned from implementing the parenting Program P in Bolivia to prevent family violence. Eval Program Plann. 2023 Apr;97:102207. doi: 10.1016/j.evalprogplan.2022.102207. Epub 2022 Dec 21. PMID: 3658743 2. Part of the reason for this was not doing good formative research or meeting men where they are; for instance men worked long hours and commuted so had limited time available for the time and location of the parenting programme sessions. I think this can come out too more in the recommendations of the importance of engaging men where they are and trying to address barriers to their engagement/meet their particular needs.

Thank you for sharing this relevant paper. We have added discussion of it in the introduction on p 5, lines 104-105: “It can also be difficult for men who work long hours to travel long distances for programming.”

We have also drawn out the potential solutions you have suggested in the discussion p 31, lines 679-681: “This is in line with previous research on engaging men in VAWG programming, which highlights the importance of providing men with flexible programming that they can engage with at a place and time that is convenient for them [26].”

In the introduction, it would also be great if you could describe evidence of additional benefits of engaging men as fathers to prevent both violence against children and IPV? See for example the Bandheberho evaluation in Rwanda: .Doyle K, Levtov RG, Barker G, Bastian GG, Bingenheimer JB, Kazimbaya S, Nzabonimpa A, Pulerwitz J, Sayinzoga F, Sharma V, Shattuck D. Gender-transformative Bandebereho couples' intervention to promote male engagement in reproductive and maternal health and violence prevention in Rwanda: Findings from a randomized controlled trial. PLoS One. 2018 Apr 4;13(4):e0192756. doi: 10.1371/journal.pone.0192756. PMID: 29617375; PMCID: PMC5884496.

We have added this on p 4, lines 90-92: “Additionally, research has found that engaging men as fathers can reduce physical and sexual violence against their partners, and physical punishment of children [17], including in Tanzania [18].”

A recent evidence review published by the Prevention Collaborative (2025) also indicates the value of engaging men as fathers to prevent child maltreatment and IPV, as well as other benefits, including to strengthen men's mental health, and strengthen quality of relationships with their children. I think emphasizing this is important to not only consider men as instrumentalist to the benefits of women and children (absolutely important) but also consider the benefits to men through their engagement in such programming.

We have added this on p 4, lines 92-94: “Parenting programmes have also been linked to improved parental mental health, which in turn contributes to reductions in violence and maltreatment of children and strengthened parent-child relationships [19].”

In the introduction the authors suggest that joint, or family-wide interventions might be more effective" clarify "than engaging fathers alone"; you could also link this to the benefits of accountability to women and girls through these family wide interventions.

Thanks, we have added this on p 5, lines 117-118.

On page 7, I would say 'most programmes have focused on early child outcomes' including to prevent violence against children, as many parenting programmes have violence against children as a primary outcome, and this links it back more broadly to focus on gender transformative programming and age-disparate sexual relationships.

We have added this on pp 6-7, lines 142-146.

Existing interventions targeting men as fathers of adolescent girls that I am more familiar with are also ones preventing VAC including preventing child marriage. For instance, there is an ongoing adaptation of the Indashyikirwa programme in Syria which has targeted couples who live with adolescent girls to prevent both IPV and child marriage of the parents' adolescent girls.

Thanks, the ‘Parenting for Lifelong Health’ programme in Tanzania (which you reference below) also targeted men as fathers of adolescent girls. We have added a paragraph describing this intervention in the introduction as an example.

The method are very rigorous methods and I commend the very rich findings.

Thank you for these positive comments.

In the discussion, I am not sure it's fair to say while men's engagement with parenting programmes is often low, as we have many examples (including the Bandeberho from Rwanda), or this recent scale up of a parenting programme in Tanzania, of successfully engaging men in parenting programmes. I do think it is important to reference the latter study, especially since it is of parenting programmes with adolescent girls: Jamie Lachman, Joyce Wamoyi, Mackenzie Martin, Qing Han, Francisco Antonio Calderón Alfaro, Samwel Mgunga, Esther Nydetabura, Nyasha Manjengenja, Mwita Wambura, Yulia Shenderovich - Reducing family and school-based violence at scale: a large-scale pre–post study of a parenting programme delivered to families with adolescent girls in Tanzania: BMJ Global Health 2024;9:e015472.

Thank you, we have added mention of the Tanzania study in the introduction on p 6, lines 129-140: “There have also been examples of successful parenting interventions in Tanzania. Recently, the ‘Parenting for Lifelong Health programme, which was adapted from the South African context, was found to be effective in reducing child maltreatment, IPV, and parent and child depression, as well as improving sexual health communication between parents and daughters, among other outcomes [23]. The programme was developed to address the underlying drivers of HIV among adolescent girls through a 14-session curricula which was delivered to over 75,000 parents and adolescent girls [23]. Over one-third of the parents included were men, suggesting there is a high demand for parenting programmes from male caregivers in Tanzania. To further scale-up this programme while reducing costs, the researchers developed a phone application to deliver the material to parents on mobile phones, which were distributed throughout communities in the Mwanza region [29]. This ‘ParentApp’ programme is currently being evaluated through a cluster-randomised controlled trial, which is expected to come to completion in 2025 [29].”

We have also added it as a reference in the discussion, and changed the wording from engagement is “often low” to “can at times be limited”.

I think you can definitely note many barriers to engaging men as fathers but also lots of examples of programmes that have done this. I would definitely cite Equimundo's recent report on what worked with Programming P to engage men as fathers here for examples of what has been learned about what works to engage men as fathers in parenting programmes.

Alemann, Clara, Rachel Mehaffey, and Kate Doyle. 2023. Core Elements of Gender-Transformative Fatherhood Programs to Promote Care Equality and Prevent Violence: A Practitioner Guide. Washington, DC: Equimundo. equimundo.org/resources/core-elements-of-gender-transformative-fatherhood-programs-to-promote-care-equality-and-prevent-violence/.

We have added this on p 31, lines 681-687: “Other strategies that have been identified as key to promoting men’s engagement include strong partnerships with implementing partners and in-country stakeholders, creating contextually appropriate content based on formative research, taking the time to adequately train facilitators, engaging influential community members to support normative change and securing funding from flexible donors [59]. These strategies were also adopted by LINEA and are further described in an intervention development paper by Howard-Merrill et al. [43].

In limitations: could you add something about positionality of authors/researchers to the participants and how this was navigated?

We have added this on pp 34-35, lines 769-773: “ This may be, in part, be due to participants long-standing relationship with TCRS, which fostered trust in both the organisation and the study. While the AGO researchers were more educated and came from a more urban area then the participants, being introduced by TCRS and having plenty of experience working with the community likely helped establish credibility and comfort.”

The authors' important point around accountability to women and girls is great; perhaps you can also note something around there can be limitations/risks with notion of men protecting girls including controlling behaviours or limiting agency on behalf of girls around who they are interested/in choose to date. It would be great if this is reflected on a bit more in discussion, including perhaps noting in the limitations that you do not have the perspective from adolescent girls themselves.

Thank you for raising these important points. We have added a discussion of these risks on p 34, lines 751-754: “ In supporting this change, practitioners must be careful not to inadvertently promote men’s controlling behaviours by ensuring they are respectful of the agency of adolescent girls’, including, for example those who do choose to date.”

We have also added the limitation on p34, lines 762-765: “While data from adolescent girls were included in our companion papers (see [44, 45]), findings did not speak directly to their perspectives of the role of male caregivers in preventing age-disparate transactional sex, and therefore data from girls were not included in this study.”

Reviewer 2

I have done the review and the paper is well presented, and a significant contribution to knowledge on the subject. Please see the attached minor comments for review.

Thank you for these positive comments. We look forward to addressing your comments.

Authors should consider starting a new paragraph with a strong and direct statement. Therefore, the sentence in Line 63 "Others have also questioned..." should be revised. This comment is also relevant for Line 94 and 115.

In response to this comment we have updated the beginning sentences of these paragraphs. “Others have also questioned…” has been combined with the paragraph above, so the second paragraph begins: “Evaluations of interventions aimed at engaging men…” on p 3 line 61.

We have made similar edits to the beginnings of the following paragraphs on p 5, line 98 and 119.

Concept of Transactional Sex: Authors may consider starting the section "Age-disparate transaction sex" Line 131, with basic concepts of the subject presented in later paragraphs - Line (140 -146) and part of the subsequent paragraph (Line 147 -151) before detailing its consequences (HIV and VAW)

We have chosen to keep the current ordering of these paragraphs as we think the paragraph on HIV provides a nice segway from the previous section to discussing age-disparate transactional sex.

Line 162 - It notes that the evaluation was carried out in two regions of Tanzania - Kigoma and Shinyanga. I have not seen in the paper the rationale to exclude Kigoma results or focusing in Shinyanga only

We have added a sentence on p 9, lines 201-204 describing why we present the findings from Shinyanga and Kigoma separately, and where the findings from Kigoma have been published. It reads: “Findings from the study conducted in Kigoma region, where the radio drama was broadcast on a local radio station rather than given to participants on a flash drive, are presented in another paper by Howard-Merrill et al. [54].”

Contextualization - I lack context specific information of the problem in Tanzania and Shinyanga specifically in regard to age-disparate sex, VAWG and men's involvement as agent for change and care givers. I assume the context may have been described elsewhere as the current paper is indicated to be part of three. However, readers of the current paper will benefit from a brief description of the same

In response to this comment we have added a brief description of Shinyanga region on p 8, lines 180-194: “We selected Kishapu district in the Shinyanga region as the study site for several reasons, including ease of implementation and the prevalence of issues associated with age-disparate transactional sex. The choice was also influenced by a pre-existing relationship with the implementing partner organisation, the Tanganyika Christian Refugee Service (TCRS), which is based in the area. TCRS programming focuses primarily on building community resilience for climate change adaptation and mitigation, and preventing gender-based violence. Shinyanga is a predominantly rural region where agriculture (crop cultivation and livestock rearing) is the main occupation [46]. It has higher than average rates of HIV [47] compared to the national average (4.4%) [48], and among the highest prevalence of IPV in the country [49], with estimates as high as 78% of ever-married women reporting experiencing violence in their lifetime [50]. The region also has the highest prevalence of child marriage in Tanzania, with estimates between 59-64% [50, 51], and a recent study highlighting a culture of tolerance towards sexual violence among married adolescent girls [52]. However, research with caregivers and daughters has also shown strong support for parent-child connectedness as a strategy to prevent early sexual debut among adolescent girls [53].”

Methodology

The radio drama "Msichana wa Kati" that was designed to be played in full for nine months (Line 194 - 196) - when did it start? Did it run for nine months as required? On the other hand, authors indicate that baseline evaluation (conducted before the intervention) and endline were 2 -3 months apart (Line 183 -184). There is a date mention in Line 2

---

## [Editor Report · Decision Letter 2]

3 Mar 2025

“The child of your fellow is your child”: Building on existing protective norms to engage men as caregivers; qualitative findings from an exploratory evaluation of an edutainment intervention to prevent age-disparate transactional sex

PONE-D-24-32227R2

Dear Dr. Pichon,

Thank you for your prompt and thorough response to the request for revisions.  We’re pleased to inform you that your manuscript has been judged scientifically suitable for publication and will be formally accepted for publication once it meets all outstanding technical requirements.

Kind regards,

Jessica Leight, PhD

Academic Editor

PLOS ONE
---

## [Editor Report · Acceptance letter]

PONE-D-24-32227R2

PLOS ONE

Dear Dr. Pichon,

I'm pleased to inform you that your manuscript has been deemed suitable for publication in PLOS ONE. Congratulations! Your manuscript is now being handed over to our production team.

Kind regards,

on behalf of

Dr. Jessica Leight

Academic Editor

PLOS ONE